# Metabolic Profiles and Blood Biomarkers to Discriminate between Benign Thyroid Nodules and Papillary Carcinoma, Based on UHPLC-QTOF-ESI^+^-MS Analysis

**DOI:** 10.3390/ijms25063495

**Published:** 2024-03-20

**Authors:** Gabriela Maria Berinde, Andreea Iulia Socaciu, Mihai Adrian Socaciu, Gabriel Emil Petre, Carmen Socaciu, Doina Piciu

**Affiliations:** 1Department of Occupational Health, University of Medicine and Pharmacy “Iuliu Haţieganu”, Str. Victor Babes 8, 400347 Cluj-Napoca, Romania; gabriela_berinde@yahoo.com; 2Department of Medical Imaging, University of Medicine and Pharmacy “Iuliu Haţieganu”, 400162 Cluj-Napoca, Romania; mihai.socaciu@umfcluj.ro; 3Department of Surgery 4, University of Medicine and Pharmacy “Iuliu Hatieganu”, 400489 Cluj-Napoca, Romania; dr_gabipetre@yahoo.com; 4Research Center for Applied Biotechnology and Molecular Therapy BioDiatech, SC Proplanta SRL, Str. Trifoiului 12G, 400478 Cluj-Napoca, Romania; csocaciu@proplanta.ro; 5Doctoral School, University of Medicine and Pharmacy “Iuliu Haţieganu”, 400012 Cluj-Napoca, Romania

**Keywords:** papillary thyroid cancer, benign nodules, serum metabolomics, oncometabolites, biomarker and pathway analysis

## Abstract

In this study, serum metabolic profiling of patients diagnosed with papillary thyroid carcinoma (PTC) and benign thyroid pathologies (BT) aimed to identify specific biomarkers and altered pathways when compared with healthy controls (C). The blood was collected after a histological confirmation from PTC (n = 24) and BT patients (n = 31) in parallel with healthy controls (n = 81). The untargeted metabolomics protocol was applied by UHPLC-QTOF-ESI^+^-MS analysis and the statistical analysis was performed using the MetaboAnalyst 5.0 platform. The partial least squares-discrimination analysis, including VIP values, random forest graphs, and heatmaps (*p* < 0.05), was complemented with biomarker analysis (with AUROC ranking) and pathway analysis, suggesting a model for abnormal metabolic pathways in PTC and BT based on 166 identified metabolites. There were 11 classes of putative biomarkers selected that were involved in altered metabolic pathways, e.g., polar molecules (amino acids and glycolysis metabolites, purines and pyrimidines, and selenium complexes) and lipids including free fatty acids, bile acids, acylated carnitines, corticosteroids, prostaglandins, and phospholipids. Specific biomarkers of discrimination were identified in each class of metabolites and upregulated or downregulated comparative to controls, PTC group, and BT group. The lipidomic window was revealed to be more relevant for finding biomarkers related to thyroid carcinoma or benign thyroid nodules, since our study reflected a stronger involvement of lipids and selenium-related molecules in metabolic discrimination.

## 1. Introduction

Thyroid diseases have had an increasing incidence all over the world, being characterized by a wide range of physical and mental symptoms that can affect biological functions as well as the emotional and social lives of people. Numerous occupational and environmental risk factors have been shown to disrupt the endocrine function and to be related to different pathologies, including benign thyroid nodules (BT) and thyroid carcinoma, the most common type of endocrine-related cancer [1,2,3,4,5,6,7,8]. 

The advances in the diagnosis and therapy of thyroid cancer (TC) were recently reviewed [9], including the management guidelines for patients with benign thyroid nodules and differentiated thyroid cancer [10], with papillary carcinoma (PTC) being the most common type of thyroid cancer, ranked ninth in the Globocan 2020 top ten [11].

Nodular thyroid pathologies are often diagnosed by routine ultrasound examinations, fine-needle aspiration as a diagnostic gold standard, and a histological examination after surgery [12,13,14]. 

Clinically, the diagnosis of thyroid dysfunction is primarily based on biochemical indicators, such as serum-thyroid-stimulating hormone (TSH) and serum-free thyroxine (FT4). The diagnostic criteria for hypothyroidism include higher serum TSH and serum FT4 levels below the reference range. PTC is diagnosed frequently in adults (30–50 years old), especially in women. The current knowledge of the etiology of PTC remains limited, but Proline/Proline genotypes might have an increased risk of developing PTC and the autoimmune disorder Hashimoto’s disease after adjusting for gender, age, smoking, alcohol, and drug consumption [15]. Up to 50% of the new cases of the thyroid carcinoma group are papillary microcarcinoma (mC), with nodules less than 1 cm.

Until now, no reliable and specific diagnostic molecular markers for the detection and staging of thyroid cancer have been standardized; the assessment of metabolic changes in the development of thyroid nodules or cancer is limited to individual hormones and some metabolite levels by standard clinical tests. In this context, new, reliable, and affordable biomarkers for the detection and accurate diagnosis of thyroid diseases are needed to complete and improve current methods. 

The “omics” technologies address distinct directions for diagnosis and investigation, from genomics to proteomics and metabolomics. Recent reviews have shown diverse techniques applied for an accurate molecular diagnosis, including genomic PCR, proteomics by MALDI-TOF–MS, and serum profiling for identifying potential biomarkers using HPLC–MS data [16]. MicroRNA, DNA mutations, and circulating cells targets were investigated as potential biomarkers [17]; in addition, thyroglobulin (Tg) and its antibody have been recommended as post-thyroidectomy biomarkers for TC surveillance [18].

Metabolomics is an emerging technology which can separate, identify, and classify different classes of small molecules (<5000 Da), including metabolites that may indicate the “downstream” effects of altered metabolic pathways in benign or malignant transformation [19,20,21]. Some metabolic alterations in TC have been identified by techniques based on high-performance gas or liquid chromatography coupled with mass spectrometry (GC–MS and LC–MS) or NMR [9,11,18,22,23,24,25,26,27,28,29].

In the last decade, a growing interest in finding specific metabolite alterations in tissue and plasma samples in TC patients compared with benign nodules has been reported, e.g., tissue oncometabolites (higher lactate and choline levels and low levels of citrate, variations of glutamine and glutamate levels being considered putative biomarkers for BTs). Lipids, such as cholesterol, phosphocholines, and derivatives have shown significant alterations in TC patients vs. healthy subjects [26]. The distribution of phosphatidylcholines (16:0/18:1; 16:0/18:2) and sphingomyelin (d18:0/16:1) is significantly higher in PTC [26].

In the last decade, systematic reviews regarding the proteomics and metabolomics techniques for studying thyroid diseases (especially PTC) have been published [24,25,26,27,29,30,31]. Elevated tissue levels of some amino acids, as well as the involvement of galactose metabolism and polar lipids, have also been noticed [32,33].

Currently, in spite of the progress made in thyroid metabolomics, there are just a few studies, in clinical research, related to thyroid metabolic profiles using biofluids (blood or urine). Hence, this study aimed to perform a comparative analysis of the serum metabolic profiles of PTC and BT patients in comparison to healthy controls (1), to identify the specific putative biomarkers for differentiating between malignant and benign groups (PTC vs. BT) based on metabolomics analysis coupled with multivariate statistics (2), as well as to identify the specific classes of metabolites responsible for the alteration of metabolic pathways in benign vs. malign thyroid pathology (3), in order to provide a reliable metabolic overview of the thyroid pathophysiology.

## 2. Results

### 2.1. Metabolomic Profiles to Differentiate Metabolites between PTC, BT, and C Group Samples

According to the raw data obtained by the UHPLC-TOF-ESI^+^-MS analytical procedures (see Section 4), from an initial number of 386 molecules identified, 166 metabolites belonging to 10 different metabolites classes were selected, as presented in Appendix A. The partial least squares-discriminant analysis (PLSDA), variable importance in the projection (VIP) scores, heatmap, random forest (RF), and biomarker analysis were used as statistical algorithms to evaluate the significance of differences between the three groups of subjects.

According to the PLSDA score plots, a significant difference between groups C and PTC and BT (Figure 1A), with a covariance of 32.1%, was obtained. The cross-validation analysis showed a good accuracy (0.8) with R2 values > 0.6 and Q2 values > 0.5 for the first three components, confirming an acceptable predictability and reliability of the model. The VIP scores > 1.5 revealed the first 15 molecules to be considered significant for the discrimination between these groups. The MDA values > 0.0008, according to RF analysis, showed the most relevant molecules to be considered as putative biomarkers for differentiation (Appendix A). The cross-validation plot derived from the PLSDA analysis had a high accuracy, with high R2 values (>0.9) and a significant Q2 value (>0.9) for the first component, confirming a good predictability of the model. Considering the VIP values > 2, and MDA values > 0.010, the heatmap (Figure 1B) illustrates a number of 16 molecules with decreased levels in PTC and BT groups comparative to controls, and 9 molecules (orange-brown colored) with increased levels in PTC and BT (*p* < 0.05). For the identification report, see Appendix A.

A similar PLSDA analysis was applied to discriminate between the groups PTC vs. C and PTC vs. BT, as presented in Figure 2A,C, respectively.

The PLSDA score plots showed a significant discrimination between PTC and C (Figure 2A); the heatmap (Figure 2B) showed 12 molecules with increased levels in PTC vs. C and 13 molecules with decreased levels in PTC, when a threshold of *p* < 0.05 was considered. The VIP scores were >2 only for selenomethionine (*m*/*z* = 196.9180), and VIP scores were >1.5 for eight other molecules (Appendix A) with decreased levels in the PTC group comparative to Controls. The cross-validation plot had a high accuracy, with high R2 values (>0.8) and a significant Q2 value (>0.7) for the first component, confirming a good predictability for the model.

Figure 2C,D show, through a similar analysis, the discriminations between PTC and BT groups. The PLSDA score plot shows an acceptable discrimination between the groups (covariance of 22%) (Figure 2C), with seven molecules with decreased levels in PTC group according to VIP > 1.5 (Appendix A). The heatmap (Figure 2D) shows 12 molecules with decreased levels in PTC vs. BT and 13 molecules with increased levels in PTC (*p* < 0.05). Considering all the previous algorithms, the molecules selected as potential biomarkers to discriminate the PTC from BT are presented in Table 1.

### 2.2. Biomarker Analysis

According to Metaboanalyst 5.0, the algorithm for biomarker analysis was applied to obtain the receiver operating characteristic (ROC) curves in order to evaluate the diagnostic power of a biomarker and the area under the curve (AUROC), which provides values characterized by a higher sensitivity vs. specificity for biomarkers. The metabolites with the highest AUROC values were considered the best biomarkers for differentiation between the groups PTC vs. C and PTC vs. BT (Table 2). The log2FC values calculated below indicate an increased (negative values) or decreased (positive values) level of the metabolite in PTC group vs. C or BT group, respectively.

Many of these molecules were confirmed as potential biomarkers also by complementary statistics, as shown in Section 2.1. According to the AUROC values, the discrimination between PTC and C groups was higher (AUROC values over 0.966 among the 17 molecules ranked) than for PTC vs. BT (AUROC values ranging from 0.730 to 0.644) in this cohort of selected molecules.

### 2.3. Pathway Analysis

The cohort of molecules separated and identified in all groups were subjected to pathway analysis considering the matched metabolic pathways, according to the *p*-values obtained from pathway enrichment analysis and to the impact values (ranging from 0.16 to 1) (Figure 3). A higher value on the y-axis indicates a lower *p*-value (threshold *p* < 0.05), while the x-axis gives the pathway impact value. The dimension and color of the circles in the graphic illustrate the most important first 10 metabolic pathways, and the list below includes the first 19 altered metabolic pathways listed in a decreasing order based on their impact values.

### 2.4. Differential Statistics for Each Class of Molecules Involved in the Thyroid Pathology

From the cohort of molecules selected for statistical analysis, different classes of molecules were identified, according to their involvement in specific metabolic pathways (Appendix A). For each class, the matrices (*m*/*z* values vs. peak intensities) were submitted to MetaboAnalyst analysis in order to identify, more specifically, the molecules which may differentiate between groups PTC and BT, vs. C. The PLSDA score plots and VIP scores for this analysis are represented in Appendix A. Figure 4 illustrates the heatmaps obtained by applying the Ward clustering method, with Euclidean distance and ANOVA *p*-value (FDR) cutoff of 0.05.

The data are consistent with the results of pathway analysis (see Section 2.3) as they show details regarding specific variations of different classes of metabolites, identifying either common metabolic signatures of PTC and BT or specific alterations for each pathology group, compared to controls. These findings are consistent with other previous experimental data obtained by LC–MS or NMR analysis of serum PTC and/or BT (hypo- and hyperthyroidism) samples that also confirmed the presence of specific alterations of these metabolic pathways [24,25,34,35,36,37,38,39,40,41].

Regarding the polar metabolite-related pathways, the heatmaps illustrate similar patterns for PTC and BT groups, with some exceptions, as seen in Figure 4A–D. We identified alterations in the TCA and amino acid metabolism by decreased levels of citric, fumaric, succinic acids, except glutamine, glutamic acid, histidine, glycine proline, proline betaine, hydroxylysine, and phenylalanine (1), decreased levels of hydroxybutyric acids and GABA, decreased levels of selenomethionine and selenocysteine (precursor of glutathione), and an increased level of homocysteine (2). Tryptophan and kynurenine, which are actively metabolized in tumors through the kynurenine pathway, showed reduced serum levels in both PTC and BT samples. Taurine, hypotaurine, tyramine, and methyl histidine had opposite behaviors and were able to differentiate between PTC and BT groups (3). Alterations of purine, pyrimidine, nucleotide, and nucleoside levels were also identified, having decreased levels, especially the adenine- and guanine-derived metabolites. From this category, methyl guanosine seems to act as a good marker to differentiate between PTC and BT groups (4).

Regarding the lipid metabolites (Figure 4E–J), the heatmaps illustrate more important differences between PTC and BT groups, except for the corticosteroids category. The levels of saturated FFAs were increased in the PTC group, where the levels of unsaturated FFAs were decreased. The BT group had the highest levels of short-chain FFAs (C14:0, C16:0, and C16:1). The PTC group had the highest levels of oleic acid derivatives (C18:1, C18:2), metabolites through which the PTC group can be well differentiated from the BT group (5). Out of the 22 acylcarnitine identified, the acylated carnitines with the longest chain had increased levels in PTC and BT groups, and 6 specimens had increased levels only in the PTC group, being actively involved in the transport of acyl groups from fatty acids to mitochondria (6). Bile acid metabolism seems to be affected by increased levels of deoxycholic acid derivatives, especially in the BT groups and also by decreased levels of different cholic acids in the PTC group (7). Phospholipids, especially lysoPC, lysoPE, and lysoPA, showed variations in the three sampled groups, some with significant increases in the PTC group, especially the unsaturated acids C22:6, C20:4, C18:2, and 16:1 (8). Corticosteroids (cortisone and cortisol derivatives) showed increased levels (except for tetrahydrocortisol) in the PTC and BT groups compared to controls. Dihydrocortisol was able to discriminate PTC vs. BT (9). The prostaglandins A and E discriminated well the PTC and BT groups, with increased levels of PGE2 and pGA1 in the BT group compared to the PTC group, while PGA2/B2 had increased levels in the PTC group and decreased levels in the BT group (10).

## 3. Discussion

In this study, the serum metabolic alterations in the thyroid pathology groups (PTC and BT) compared to the control group (C) were investigated. We identified different classes of metabolites corresponding to specific pathways that are able to differentiate between the three groups. During tumor development, the intensification of tumor glycolytic activity, the amino acid turnover for protein synthesis, the increased production of metabolic substrates, and precursors needed to increase the energy needed for cell proliferation are well documented. Therefore, higher serum levels of such precursors are common for PTC and BT. Reduced levels of citric acid are explained by its conversion to acetyl-CoA, needed for fatty acid biosynthesis, while the decreased levels of other amino acids that we identified were due to their conversion to proteins. In the PTC group, the proline and glutamic acid derivatives levels were elevated, and the levels of hydroxy butyric and dihydroxy butyric acids, that represent intermediates of fatty acid metabolism, were lower compared to BT and C groups. Similar data were reported also by other authors [32] who identified, by GC–MS analysis, putative circulating biomarkers such as sucrose, cysteine, cystine, glutamic acid, a-ketoglutarate, 3-hydroxy butyric acid, purine and pyrimidine metabolites, and adenosine-5-monophosphate, as well as metabolites of the fatty acid metabolism that can be used for the differential diagnosis between malignancy and benignity in thyroid nodules. Also, by UPLC-QTOF-MS applied for serum samples, increased levels of hydroxy butyric acid, docosahexaenoic, and 1-Meadenosine were noticed [37], as well as alterations of 42 metabolites, including proline betaine, and decreased levels of Lyso-phospholipids PC (18:0) and (18:1) [29]. A large study performed on serum metabolomics that compared PTC patients with healthy subjects also identified biomarkers such as proline betaine, taurocholic acid, L-phenylalanine, retinyl beta-glucuronide, alpha-tocotrienol, threonine acid, L-tyrosine, L-tryptophan, 2-arachidonylglycerol, and citric acid as being upregulated in the PTC group, while other 42 metabolites were downregulated in this group [22]. This study also identified six abnormal metabolic pathways, 3-hydroxy-cis-5-tetradecenoylcarnitine, aspartyl phenylalanine, L-kynurenine, methylmalonic acid, phenylalanyl phenylalanine, and L-glutamic acid, with AUC values > 0.75, pathways related to the differentiation of metabolites, which could be involved in the pathophysiology of PTC. The Warburg effect was reflected in PTC by the levels of 3-hydroxy-cis-5-tetradecenoylcarnitine, aspartyl phenylalanine, L-kynurenine, methylmalonic acid, phenylalanine, and L-glutamic acid, concluding that the aspartic acid and glutamic acid metabolism, the urea cycle, and the tricarboxylic acid cycle were involved in the PTC pathogenesis [22].

A comparison between PTC and nodular goiters patients vs. healthy controls, based on LC–LTQ orbitrap mass spectrometry, acknowledged differences in the levels of 3-hydroxy butyric acid (an intermediate in the fatty acid metabolism) in PTC and control groups, as well as differences in the C16/C2/L-Carnitines ratios and in the levels of Sphingosine and Sphingosine-1-phosphate, indicating that these molecules are potential diagnostic markers. In this study, it was concluded that the PTC and BT metabolic profiles were both similar and remarkably different, by mechanistic pathways [38].

Circulating free fatty acids are important for energy supplementation when needed. Different levels of FFAs were found in the serum samples of BT and PTC patients. The benign thyroid pathology seems to be associated with an accelerated FFA metabolism, while the lower levels of FFAs in the PTC group can be associated with an increased demand of lipids from tumor cells. In addition, increased levels of bile acids were correlated with the upregulated FFA metabolism, which was observed in the BT group. These data are confirmed by the findings of different study groups [24,25,33,34,35,36,37,38,39,40,41,42,43,44,45,46], their results being related to the same categories of metabolites.

The benign nodular goiter can be associated mainly with hypothyroidism. Recently, a metabolomic study revealed systemic metabolic alterations of subclinical and clinical hypothyroidism on primary bile acid and steroid hormone biosynthesis, lysine, tryptophan, and purine metabolism. Seventeen biomarkers were validated, with significant associations with thyrotropin, FT4, thyroid peroxidase antibody, or thyroglobulin antibody levels [39].

The oncometabolites identified by untargeted and targeted omics analysis revealed that several molecules, e.g., glucose, fructose, galactose, mannose, rhamnose, 2-keto-gluconic and malonic acids, inosine, citrate, lactate, several amino acids, purine and pyrimidine metabolites, fatty acids, cholesterol and arachidonic acid, choline-derived phospholipids, citrate, and lactate are the most significant biomarkers that can be applied to differentiate between benign and malignant thyroid diseases [27].

Elevated tissue concentrations of some amino acids (methionine, leucine, tyrosine, and lysine), polar lipids (especially phospholipids and sphingolipids), and de novo synthesis of fatty acids have been identified in PTC serum by GC–MS [32]. By untargeted GC-TOF–MS, a preliminary screening of potential biomarkers in PTC tissues was performed and the galactose metabolism pathway was identified as important, influencing PTC development, affecting the energetic metabolism [33].

The metabolomic profile of patients with follicular tumor vs. follicular nodules was analyzed and compared by LC–MS. The analysis identified six types of lipids, which may explain the discrimination between the metabolic profiles. In this category were included amino acids (L-glutamate, L-glutamine), Lyso-PA, and Lyso-PC derivates of many fatty acids, mainly C16, C18, C20, and C22, but also sphingomyelin (d18:0/12:0) and linoleic acid. Using KEGG and HMDB databases, nine metabolic pathways were found, including four amino-acid-related metabolic pathways and one lipid metabolic pathway. The abnormal levels of LysoPA may be related to follicular tumor carcinogenesis, caused by a lipid metabolic pathway abnormality [25].

Increased attention is attributed to selenium and its role in cancer prevention and promotion, as documented by many authors [41,42,43,44]. Selenium acts mainly through selenoprotein complexes, such as oxidoreductases involved in glutathione-dependent hydroperoxide removal, the reduction in thioredoxins, the selenophosphate synthesis, the activation and inactivation of thyroid hormones, the thioredox-independent repair of oxidized methionine residues, and in the ER-associated thyroid hormone deiodinases, activating or inactivating their degradation [42]. Selenium is critical for the function of the thyroid gland, where it is particularly abundant, as selenocysteine is in the active site of the three peroxidases, and its deficiency is a very common condition nowadays. The role of selenium and selenoproteins in the thyroid pathophysiology (e.g., autoimmune Hashimoto’s thyroiditis and Graves’ disease) was recently reported [43], with a major frequency in middle-aged women. It is known that patients with hypothyroidism, subclinical hypothyroidism, autoimmune thyroiditis, and enlarged thyroid have reduced selenium levels.

The role of selenium in the pathophysiology of the thyroid gland is well documented, as significant correlation between low selenium serum levels, nodules development, carcinoma progression, and variations of selenium in different pathological groups (from PTC to Hashimoto or Goiter nodules) were recently reviewed [44]. Selenomethionine is known to have a vegetable origin but can also be synthesized in yeast, while selenocysteine is of animal origin being incorporated into iodothyronine deiodinases. Both compounds play an essential role in the metabolism of the thyroid hormones [44]. A recent meta-analysis clarified the association of selenium, copper, and magnesium levels with thyroid cancer, concluding that patients with thyroid cancer had lower levels of serum selenium and magnesium vs. higher copper levels, when compared with healthy controls [45]. Such an association seems to be correlated with the antioxidant properties of the selenoenzymes, relevant in carcinogenesis and tumor progression, although the specific mechanisms are not yet fully understood. Although the selenium levels were not significantly lower in patients with thyroid cancer, serum selenium levels were inversely correlated with the cancer stage [46]. Therefore, our data showing decreased levels of selenomethionine and methylselenocysteine in both PTC and BT groups compared to controls are in good agreement with the above-mentioned study findings.

## 4. Materials and Methods

### 4.1. Patients and Study Design

This study complied with the guidelines of the Declaration of Helsinki and the Conference for Coordination of Clinical Practice and was approved by the Ethics Committee for Scientific Research (DEP224/26 July 2022) of the “Iuliu Hatieganu” University of Medicine and Pharmacy Cluj-Napoca, Romania. The written informed consent was obtained from all subjects. We had a total number of 81 healthy subjects (group C) and 55 patients diagnosed with different thyroid pathologies. Patients diagnosed with papillary carcinoma (CA) and micropapillary carcinoma (mC) were included in the PTC group (n = 24), and patients with benign thyroid diseases, such as nodular goiter (G), Hashimoto’s thyroiditis, Bethesda II thyroiditis, and follicular adenoma, were included in the BT group (n = 31), as shown in Table 1. The patients’ data, including blood samples, were collected between October 2020 and February 2022 from a University Hospital in Cluj-Napoca, Romania. The low number of subjects included in this study could be explained by the fact that the time frame for the study was short, as it is included in a doctoral project from the “Iuliu Hatieganu” University of Medicine and Pharmacy Cluj-Napoca, Romania, the hospital is a general surgery hospital, where we had one team member and collaborating investigator, and also because the study was conducted during the COVID-19 pandemic, which highly interfered with patient presentation for medical treatment. Each participant completed a questionnaire by direct interview with the main investigator (the Ph.D. student). We collected information on the type of thyroid disease (PTC and BT), with TNM staging for PTC patients, associated diseases, information included in the personal medical history, demographic data, family history, previous medication, lifestyle (tobacco smoking, alcohol consumption, nutritional status), environmental, and occupational risk factors, which will be the topic of a different project.

The data for the inclusion, exclusion criteria, and for the risk factor assessment were collected from the patient’s clinical history and pathology reports.

The inclusion criteria consisted of history of thyroid pathology (PTC ant BT) or no previous history of thyroid pathology (C) using the diagnostic criteria for thyroid diseases and histological classification, clear consciousness, no intellectual impairment, and normal communication abilities, as presented in Table 3.

The exclusion criteria were patients that refused to participate in this study, pregnant or lactating women, combined malignancy, patients with psychiatric disorders, incomplete data for diagnostic criteria, and inconclusive pathology findings.

### 4.2. Sample Collection and Preparation

Blood was collected by venipuncture in sterile vacutainers without anticoagulant and the serum was stored at −80 °C until analysis. The samples were labeled using confidential numerical codes.

A volume of 0.8 mL mix of pure HPLC-grade methanol and acetonitrile (2:1 *v/v*) was added for each 0.2 mL of serum. For all the samples, the mixture was vortexed to precipitate proteins, ultrasonicated for 5 min, and stored for 24 h at −20 °C. The supernatant, collected after centrifugation at 12,500 rpm for 10 min (4 °C), was filtered through nylon filters (0.25 μm), introduced in glass microvials, and placed in the autosampler of the ultra-high-performance liquid chromatograph (UHPLC) just before injection. Quality control (QC) samples were also prepared in the same phase; from each sample, we collected 10 μL that were added to 2 mL Eppendorf (Eppendorf Biotech Company, Hamburg, Germany) microtubes, vortexed, and divided into 0.2 mL for each tube and pretreated using the same procedure to improve the data quality for metabolic profiling.

### 4.3. UHPLC-QTOF-ESI^+^-MS Analysis

The metabolomic profiling was performed by UHPLC coupled with electrospray ionization-quadrupole-time of flight–mass spectrometry (UHPLC-QTOF-ESI^+^-MS) using a ThermoFisher Scientific UHPLC Ultimate 3000 instrument equipped with a quaternary pump, Dionex delivery system, and MS detection equipment with MaXis Impact (Bruker Daltonics, Billerica, Mass., USA). The metabolites were separated on an Acclaim C18 column (5 μm, 2.1 × 100 mm, pore size of 30 nm (Thermo Fisher Scientific, Waltham, Mass., USA) at 28 °C. The mobile phase consisted of 0.1% formic acid diluted in water (A) and 0.1% formic acid in acetonitrile (B). The elution time was set for 20 min. The flow rate was set at 0.3 mL·min^−1^ for serum samples. The gradient for serum samples was 90 to 85% A (0–3 min), 85–50% A (3–6 min), 50–30% (6–8 min), and 30–5% (8–12 min), increased to 90% at min 20. The volume of injected extract was 5 mL and the column temperature was set at 25 °C. Several QC samples obtained from each group were used also to calibrate the separations. Doxorubicin hydrochloride (*m*/*z* = 581.3209) solution 0.5 mg/mL was added to QC samples and used as internal standard. The MS parameters that were applied for this analysis were ionization mode positive (ESI^+^), MS calibration with Natrium formate, a capillary voltage of 3500 V, the pressure for the nebulizing gas set at 2.8 Barr, drying gas flow 12 L/min, and drying temperature 300 °C. The range for the *m*/*z* values to be separated was set between 60 and 600 Daltons. The control of the instrument and the data processing was performed using the specific software TofControl 3.2, HyStar 3.2, Data Analysis 4.2 (Bruker, Daltonics, Billerica, Mass., USA), and Chromeleon 7.2.10 MUb-d, respectively.

### 4.4. Statistical Analysis

Subsequently to the sampling using UHPLC-QTOF-ESI+-MS analysis, up to 850 molecules were identified. The raw data consisted, for each sample, of base peak chromatograms (BPCs) representing the intensity of each molecule versus the retention time (min). After the sampling was finalized, a matrix was completed, cumulating all samples. The methodology applied, step by step, is presented in the Appendix A. For identification and statistical analysis, in a first step, the MS peaks having retention times under 0.8 min, intensities below 1000 units, S/N values < 5, and *m*/*z* values above 780 Daltons were removed. In a second step, the alignment of *m*/*z* values was performed using the online software www.bioinformatica.isa.cnr.it/NEAPOLIS, with the common molecules found in more than 70% of the samples being kept. After using the procedures mentioned above, a final number of 397 molecules were identified and considered for multivariate and univariate analysis. The matrices for 166 polar and lipid metabolites, including their *m*/*z* values vs. peak intensity were statistically analyzed using the Metaboanalyst 5.0 platform (https://www.metaboanalyst.ca/MetaboAnalyst/, accessed on 2 October 2023). The significant metabolic pattern was identified using international databases such as Human Metabolome Database (HMDB) and LipidMaps. The experimental *m*/*z* values were compared with the average of theoretical *m*/*z* values from the international database HMDB (Human Metabolomic Database). The accuracy of *m*/*z* values (theoretical–experimental) was below 20 ppm *(*Appendix A).

In a first attempt, the analysis was focused on the untargeted, multivariate analysis of the detected molecules identified in group C vs. PTC and BT. The supervised discriminations between groups PTC, BT, and C were determined by partial least squares-discriminant analysis (PLSDA) and random forest (RF)-based prediction test, and illustrated by heatmap clusters and correlations. The values of variable importance in the projection (VIP), as well as mean decrease accuracy (MDA), were calculated, and the ranking of the most significant molecules that explained the discrimination was achieved.

According to biomarker analysis algorithm, the ROC curves were obtained, and the AUC (area under the curve) was determined as the best prediction for differentiation of putative biomarkers. The pathway analysis was also applied based on the cohort of 166 identified molecules.

## 5. Conclusions

HPLC-QTOF-ESI^+^-MS technology may differentiate patients with thyroid pathology (maligant or benign) from healthy subjects. We identified specific putative biomarkers from different classes of metabolites, and altered metabolic pathways that were able to discriminate between PTC and BT, upregulated or downregulated in comparison to controls. The lipidomic category proved to be the most relevant for identifying biomarkers related to thyroid carcinoma or benign nodules, since our study reflected a stronger involvement of lipids and selenium-related molecules for metabolic discrimination.

Our results bring new added value through information regarding the potential use of specific putative biomarkers for differentiation between malignant, benign thyroid pathology, and healthy subjects, as presented in the Results section. The metabolic profiles of patients in the PTC group were altered due to the decrease in blood amino acids and decrease in bile acids, as well the decrease in adenosine and guanosine monophosphates levels, together with an increase in unsaturated fatty acid levels (C18:1, C18:2, C20:1).

Using a specific protocol for HPLC-QTOF-MS analysis, it is possible to identify both polar and lipid metabolites with accurate profiling via advanced statistical models offered by the online available Metaboanalyst 5.0 platform that includes different types of complementary analysis (multivariate statistics and biomarker and pathway analysis).

Further studies will be developed on larger cohorts of patients and comparative metabolomics analysis on serum versus urine samples, to improve the accuracy of biomarker diagnosis and prediction and to better identify the metabolic pathways altered in benign vs. malignant thyroid pathophysiology.

## Figures and Tables

**Figure 1 ijms-25-03495-f001:**
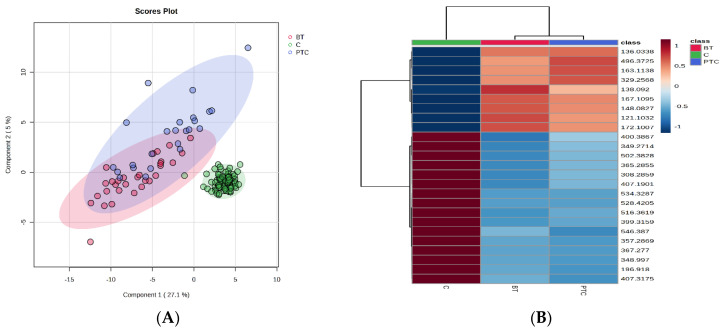
PLSDA score plot (**A**) and heatmap (**B**) for the 3 groups of patients: thyroid cancer (PTC), benign nodules (BT), and healthy controls (C).

**Figure 2 ijms-25-03495-f002:**
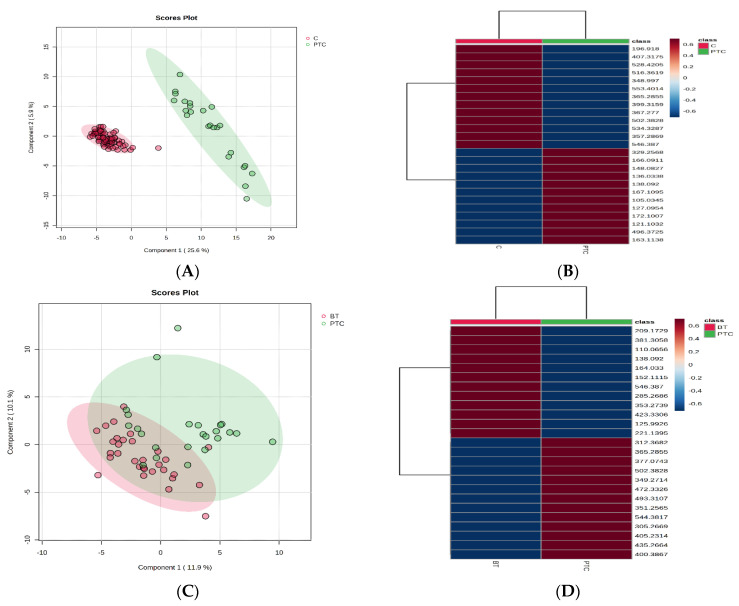
PLSDA score plots (**A**,**C**) and heatmaps (**B**,**D**) to differentiate the metabolomic profile of thyroid cancer (PTC) patients comparative to group C (control) and to BT group, respectively.

**Figure 3 ijms-25-03495-f003:**
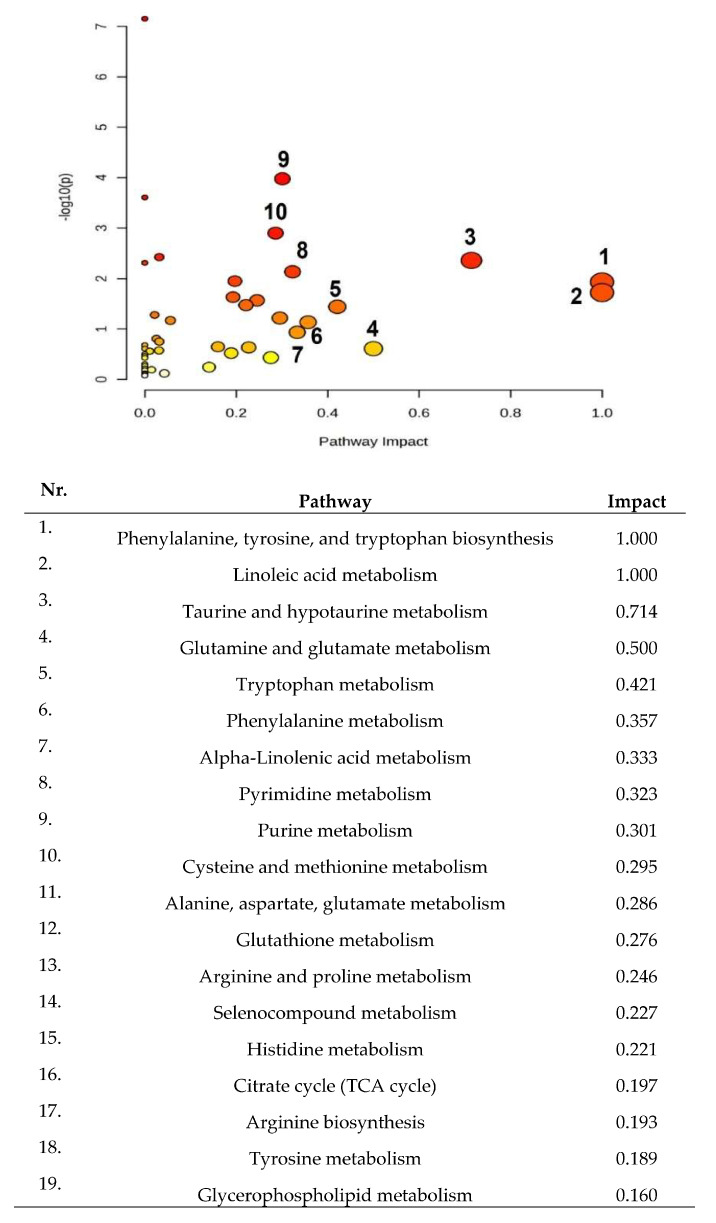
Metabolic pathways affected by thyroid diseases (PTC and BT), based on the results obtained in this study (166 molecules were selected for the statistical analysis).

**Figure 4 ijms-25-03495-f004:**
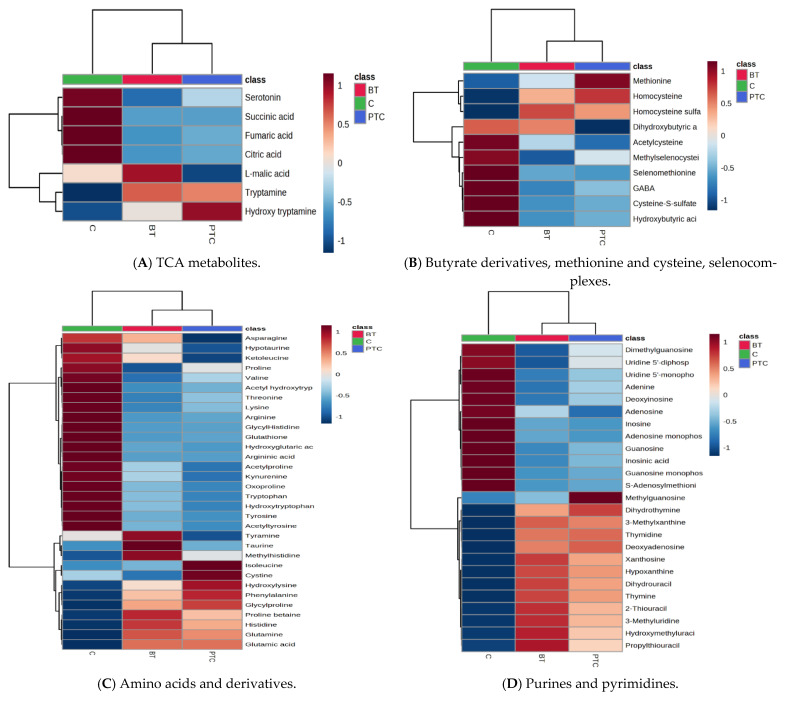
(**A**–**J**) Heatmaps corresponding to different classes of metabolites, illustrating the differences between PTC, BT, and C groups.

**Table 1 ijms-25-03495-t001:** Selection of the top 12 molecules which proved to have decreased or increased levels in the PTC group comparative to BT group of serum samples (VIP > 1, MDA > 0.002; heatmap with *p* < 0.05).

*m*/*z*	Decrease in PTC vs. BT	*m*/*z*	Increase in PTC vs. BT
110.0656	Hypotaurine	305.2669	Arachidonic acid (C20:4)
125.9926	Taurine	308.2859	Glutathione
138.0920	Tyramine	309.232	Eicosadienoic acid (C20:2)
152.1115	Hydroxy adenine	349.2714	Inosinic acid
164.0330	Acetylcysteine	351.2565	PGE3/D3
209.1729	Kynurenine	400.3867	Methyldocosanoylcarnitine (adduct)
221.1395	5-Hydroxy-L-tryptophan	405.2314	Uridine 5’-diphosphate
285.2686	Stearic acid (C18:0)	435.2664	LysoPA (18:2)
353.2739	PGE2/D2	472.3326	Cervonyl carnitine
381.3058	Dimethyl-PGE2	493.3107	LysoPA (22:1)
423.3306	7-Methyl-cholic acid	502.3828	LysoPE (20:4)
546.3870	LysoPC (20:3)	544.3817	LysoPC (20:4)

**Table 2 ijms-25-03495-t002:** Ranking of the highest 17 AUROC values of the potential biomarkers, comparative analysis of PTC group vs. controls (group C) or vs. BT group. Log 2FC values and negative or positive values indicate the intensity of relative decrease (+) or increase (−) of PTC vs. C or BT, respectively.

PTC vs. C	AUROC	log2FC	PTC vs. BT	AUROC	Log2FC
Thymine	1.000	−1.622	LysoPA (18:2)	0.730	−1.921
Methyl docosanoylcarnitine	1.000	3.012	Taurine	0.718	0.444
LysoPE (20:4)	0.999	2.066	Acetylcysteine	0.716	0.343
Taurocholic acid	0.999	2.214	LysoPE (20:4)	0.683	−1.155
Homocysteine	0.999	−3.050	Inosinic acid	0.677	−0.416
Tetrahydrocortisol	0.993	6.354	L-Palmitoylcarnitine	0.676	−0.790
LysoPE (22:5)	0.991	1.628	Cervonyl carnitine	0.676	−1.183
Histidine	0.989	−2.474	Stearic acid (C18:0)	0.668	0.407
Phenylalanine	0.988	−4.320	Uridine 5′-diphosphate	0.661	−1.325
5-hydroxylysine	0.985	−2.268	Arachidonic acid (C20:4)	0.660	−0.892
3-Methylxanthine	0.983	−1.937	PGE2/D2	0.655	0.145
PGF1a	0.980	3.086	5Hydroxytryptophan	0.651	0.156
Lithocholic acid glucuronide	0.980	1.474	7-Methyl-cholicacid	0.649	0.864
Selenomethionine	0.979	4.924	LysoPC (20:4)	0.649	−1.011
LysoPE (22:2)	0.975	2.441	Hypotaurine	0.645	0.319
3,4-Dihydroxybutyric acid	0.967	−1.122	Histidine	0.644	0.273
Dihydrocortisol	0.966	1.941	Dimethyl-PGE2	0.644	0.131

**Table 3 ijms-25-03495-t003:** Demographic and clinical data of subjects.

	Group PTC		Group BT		Controls
	CA	mC	Nodular Goiter	Thyroiditis	C
Number of participants	20	4	25	6	81
Age (years ± SD)	58.4 ± 4.0	57.6 ± 4.3	56.8 ± 4.5	58.84 ± 5.3	45 ± 8.5
Gender M/F	3/17	0/4	3/22	0/6	9/72
Body mass index (kg/m^2^)					
<30	5		10		58
≥30	19		21		23
Histologic classification (according to TNM classification)	19 Papillary carcinomas 7/pT1bN1bMx 6/pT1aNxMx 6/nonclassified 1 Medullary carcinoma/ pT3aN1b Mx	3/ pTaN0Mx 1/pT1aNxMx	0	2/Hashimoto thyroiditis 3/Thyroiditis Bethesda II 1/Follicular Adenoma (Hurtle cells)	0
TSH (mIU/L)	1.95 ± 0.92		14.66 ± 11.66		1.7 ± 0.8
FT3 (pmol/L)	5.2 ± 2.02		3.08 ± 2.52		4.8 ± 1.22
FT4 (pmol/L)	20.5 ± 8.60		12.15 ± 2.52		15.3 ± 4.8
First-degree relatives with thyroid or other cancer types					
Yes	4		3		3
No	20		28		78
Tobacco smoking (years of cigarettes)					
No	4		4		28
<5	3		6		14
5–15	12		15		35
>15	5		6		4
Alcohol consumption (total drinks/week)					
No	10/24 (41.6%)		15/31 (48.3%)		45/81(55.5%)
<5/week	8/24 (33.3%)		10/31 (32.3%)		20/81(24.7%)
>5/week	6/24 (25%)		5/31 (16.1%)		16/81(19.7%)

## Data Availability

Data are contained within the article and Appendix A.

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
