# Peer review of "Metabolic Profiles and Blood Biomarkers to Discriminate between Benign Thyroid Nodules and Papillary Carcinoma, Based on UHPLC-QTOF-ESI+-MS Analysis"

_ijms, 2024, doi:10.3390/ijms25063495_

Round 1

Reviewer 1 Report

Comments and Suggestions for Authors

The research article proves to be engaging and inclusive of essential information. Nevertheless, I have a few recommendations to enhance its current state. Some of these suggestions are outlined below:

Incorporate metabolomic profiles derived from the raw data obtained through UHPLC-TOF-ESI+ -MS. This addition will provide a more comprehensive overview and deeper insights into thyroid pathology and control group.

Clarify the rationale behind determining the number of patients in each group. An explanation of the decision-making process for patient group allocation is crucial for a better understanding. From my perspective, the limited number of patients raises concerns about the study's conclusiveness.

The conclusion lacks explicit mention of the specific metabolites that could potentially serve as biomarkers. It would be beneficial for the authors to identify and highlight these metabolites in their concluding remarks.

Since the author primarily conducted MS studies, I recommend supplementing the research with a targeted study on potential biomarkers among the metabolites. This targeted approach can offer a more focused and in-depth exploration.

Provide a precise account of the discovery of differential metabolites and the dysregulation of metabolic processes in thyroid pathology. This will contribute to a clearer understanding of the research findings.

In summary, addressing these suggestions would not only enhance the overall quality of the research article but also strengthen its scientific validity.

Comments on the Quality of English Language

 Minor editing of English language required

Author Response

Dear Reviewer 1

Thank you for evaluating our manuscript and for your relevant recommendations. Thus, these were treated accordingly. Your recommendations are valuable as these will considerably improve our manuscript.

The research article proves to be engaging and inclusive of essential information. Nevertheless, I have a few recommendations to enhance its current state. Some of these suggestions are outlined below:

1. Incorporate metabolomic profiles derived from the raw data obtained through UHPLC-TOF-ESI+ -MS. This addition will provide a more comprehensive overview and deeper insights into thyroid pathology and control group.

Answer: In the section Materials and methods, we incorporated some information regarding the main steps of the methodology applied for establishing the metabolic profiles. We included, as supplementary file this information, The methodology applied, step by step being presented in the Supplementary file S0.

  1. Clarify the rationale behind determining the number of patients in each group. An explanation of the decision-making process for patient group allocation is crucial for a better understanding. From my perspective, the limited number of patients raises concerns about the study's conclusiveness.

Answer: The low number of subjects was due to a short time frame that we had, because this study is included in a PhD project. We had also cost-related limitations, as the funding we received did not cover all the expenses. Also, we had just one collaborating surgeon, that could have provided the cases, and as he is a general surgeon, he had a low number of thyroid pathological cases that needed surgery and came back for the surgical treatment. We included just the patients for whom we had the complete set of data, including pathology reports (as we are considering in a future article to try to identify a correspondence) and also. We included an explanation in the Methods section.

  1. The conclusion lacks explicit mention of the specific metabolites that could potentially serve as biomarkers. It would be beneficial for the authors to identify and highlight these metabolites in their concluding remarks.

Answer: Please see the revised version in the Conclusion section.

  1. Since the author primarily conducted MS studies, I recommend supplementing the research with a targeted study on potential biomarkers among the metabolites. This targeted approach can offer a more focused and in-depth exploration.

Answer: Thank you! Yes, indeed, after the untargeted metabolomics and semi-targeted procedures (for classes of metabolites which showed modifications among the patients’ groups), the next step is logically, to target just the most significant biomarkers and to evaluate their levels, quantitatively.

  1. Provide a precise account of the discovery of differential metabolites and the dysregulation of metabolic processes in thyroid pathology. This will contribute to a clearer understanding of the research findings.

Answer:  The differential metabolites and the dysregulation of metabolic processes in thyroid pathology was documented until now by some reviews. Our data confirmed many of these potential biomarkers, but we brought an added value. Please see the revised version in the Conclusion section.

(This study confirmed previous data related to the specific biomarkers which may differentiate the malign vs benign pathology of thyroid. Meanwhile we found other specific, differential metabolites responsible for the dysregulation of metabolic processes in thyroid pathology, comparative to previous reports. We added in the conclusion section the new metabolites which can be considered as new biomarkers, according to this study. Please see the addition in the conclusion part (lines 477-481)).

In summary, addressing these suggestions would not only enhance the overall quality of the research article but also strengthen its scientific validity.

Answer: Thank you, your comments and suggestions are useful and surely will enhance the significance of these findings.

The modifications suggested are included in the track changes-version of the manuscript.

Thank you for your recommendations and suggestions for future research directions.

Reviewer 2 Report

Comments and Suggestions for Authors

I examined with curiosity the article titled "Metabolic profiles and blood biomarkers discriminate thyroid benign nodules and papillary thyroid carcinoma, based on UHPLC-QTOF-ESI+-MS analysis" sent to your journal. The authors collected blood after histological verification from PTC (n = 24) and BT patients (n = 31) in parallel with healthy controls (n = 81). As a result of the study, it was determined that the lipidomic window is more meaningful in finding biomarkers related to thyroid carcinoma or benign nodules, as it reflects a stronger involvement of lipids and Selenium-related molecules for metabolic discrimination. I would like to state that I found the work very valuable. My suggestion to the authors is that at least half of the introduction section can be shortened.

Sincerely

Comments on the Quality of English Language

Minor editing of English language required

Author Response

Dear Reviewer 2

Thank you for evaluating our manuscript and for your appreciation.

I examined with curiosity the article titled "Metabolic profiles and blood biomarkers discriminate thyroid benign nodules and papillary thyroid carcinoma, based on UHPLC-QTOF-ESI+-MS analysis" sent to your journal. The authors collected blood after histological verification from PTC (n = 24) and BT patients (n = 31) in parallel with healthy controls (n = 81). As a result of the study, it was determined that the lipidomic window is more meaningful in finding biomarkers related to thyroid carcinoma or benign nodules, as it reflects a stronger involvement of lipids and Selenium-related molecules for metabolic discrimination. I would like to state that I found the work very valuable. My suggestion to the authors is that at least half of the introduction section can be shortened.

Sincerely

Answer:  Thank you! We reduced the information in the Introduction section as much as possible, around 50%. Please see the modified manuscript with track changes.

Thank you for suggestions. These are valuable and will considerably improve our manuscript.

Reviewer 3 Report

Comments and Suggestions for Authors

Dear authors, 

I congratulate you for your hard work and your results regarding the serum metabolic profile of papillary carcinoma and thyroid benign nodules patients. The purpose of the study is argued, the manuscript is well composed, the statistical processing is of quality. You managed to detect specific biomarkers that differentiate between a benign and a malignant thyroid condition, even if the method needs to be validated through larger studies, on larger series of patients. 

The only advice I want to give you is correcting some minor typographical errors and arranging the manuscript in the journal's specific format: first Material and methods and then Results and Discussions.

Author Response

Dear Reviewer 3

Thank you for evaluating our manuscript and for your appreciation.

Dear authors, 

I congratulate you for your hard work and your results regarding the serum metabolic profile of papillary carcinoma and thyroid benign nodules patients. The purpose of the study is argued, the manuscript is well composed, the statistical processing is of quality. You managed to detect specific biomarkers that differentiate between a benign and a malignant thyroid condition, even if the method needs to be validated through larger studies, on larger series of patients. 

The only advice I want to give you is correcting some minor typographical errors and arranging the manuscript in the journal's specific format: first Material and methods and then Results and Discussions.

Answer:  Thank you! We revised the manuscript one more time for typing errors and we rearranged the format according to your suggestion. Please see the modified manuscript with track changes.

Thank you for suggestions. These are valuable and will considerably improve our manuscript.

Reviewer 4 Report

Comments and Suggestions for Authors *In the methodology section, additional  information about the samples taken should be included. This information should encompass an  examination of  the health history of both patients and healthy people in terms of  other disease contracted.

* Ggiven the small number of samples in each group, how are the effects of gender and age taken into account? There is significant age difference in the control group compared to the other groups (approxinately 10 years)   *In the conclusion section, specify exactly how many and which specific biomarkers were identified for the first time in this study? The studied metabolites may be used as biomarkers for differentiation, but they cannot be specific.   *In line 305 of the discussion section, the author of the article has mentioned that the obtained results have been confirmed by other people's studies. But the relevant reference numbers are not provided.

Comments on the Quality of English Language

-

Author Response

Dear Reviewer 4

Thank you for evaluating our manuscript and for your relevant recommendations.

*In the methodology section, additional information about the samples taken should be included. This information should encompass an examination of  the health history of both patients and healthy people in terms of  other disease contracted.

Answer:  We did not include much information on the health history here in this article, as this was designed for the sample analysis, but we have extensive data, which will be included in a future article. We have included just the medical history that we considered relevant for the thyroid diseases in Table 1. We have added paragraphs regarding the number of subjects in the Methodology section and the fact that the data that was retrieved by questionnaire will be the topic for another article

* Given the small number of samples in each group, how are the effects of gender and age taken into account?  There is significant age difference in the control group compared to the other groups (approximately 10 years)   

Answer:  Considering the data reported by other authors, the number of samples per group is not too low, it was the real acquisition of samples in the reported period.  The low number of subjects was due to a short time frame that we had, because this study is included in a PhD project. We had also cost-related limitations, as the funding we received did not cover all the expenses. Also, we had just one collaborating surgeon, that could have provided the cases, and as he is a general surgeon, he had a low number of thyroid pathological cases that needed surgery and came back for the surgical treatment. We included just the patients for whom we had the complete set of data, including pathology reports (as we are considering in a future article to try to identify a correspondence) and also. We included an explanation in the Methods section.

*In the conclusion section, specify exactly how many and which specific biomarkers were identified for the first time in this study? The studied metabolites may be used as biomarkers for differentiation, but they cannot be specific.  

Answer: This study confirmed previous data related to the specific biomarkers which may differentiate the malign vs benign thyroid pathology. Meanwhile we found other specific metabolites responsible for the dysregulation of the metabolic processes in thyroid pathology, comparative to previous reports. We added in the conclusion section the new metabolites which can be considered as new biomarkers, according to this study. Please see the addition in the conclusion part (lines 477-481).

 *In line 305 of the discussion section, the author of the article mentioned that the obtained results have been confirmed by other people's studies. But the relevant reference numbers are not provided.

Answer:  Thank you, in the revised manuscript we provided the references, mentioned in line 310.

Thank you for appreciating our work and for offering us suggestions for future research. These are valuable and will considerably improve our manuscript.

Round 2

Reviewer 4 Report

Comments and Suggestions for Authors

-

Comments on the Quality of English Language

Minor editing of English language required